# Cytomegalovirus Genetic Diversity and Evolution: Insights into Genotypes and Their Role in Viral Pathogenesis

**DOI:** 10.3390/pathogens14010050

**Published:** 2025-01-09

**Authors:** Cristina Venturini, Judith Breuer

**Affiliations:** Department of Infection, Immunity and Inflammation, Great Ormond Street Institute of Child Health, University College London, London WC1N 1EH, UK; j.breuer@ucl.ac.uk

**Keywords:** human cytomegalovirus, genotype, evolution, viral markers

## Abstract

Cytomegalovirus (CMV) is a ubiquitous virus that infects most of the human population and causes significant morbidity and mortality, particularly among immunocompromised individuals. Understanding CMV’s genetic diversity and evolutionary dynamics is crucial for elucidating its pathogenesis and developing effective therapeutic interventions. This review provides a comprehensive examination of CMV’s genetic diversity and evolution, focussing on the role of different genotypes in viral pathogenesis.

## 1. Introduction

Human cytomegalovirus (HCMV; species human betaherpesvirus 5) is a member of the *Betaherpesviriniae* family and is widespread among mammals, establishing a lifelong infection in its specific host. HCMV infection is common worldwide, with the prevalence of specific antibodies ranging from 60% (in developed countries) to 90% in developing countries [1].

HCMV usually causes asymptomatic infections in immunocompetent individuals; however, primary infection can result in a mononucleosis-like syndrome [2]. In addition, several studies have suggested the role of HCMV infection in the development and/or severity of inflammatory cardiovascular diseases [3], certain types of cancers [4,5], and autoimmune diseases [6]. HCMV infection can result in severe disease in immunocompromised individuals. For example, HCMV infection is a significant clinical concern in patients undergoing immunosuppressive therapy, such as solid organ and stem cell transplant recipients, and in patients with acquired immunodeficiency syndrome (AIDS) [2,7]. HCMV is the most common congenital infection in both developed and developing countries, causing sensorineural hearing loss (SNHL) and neurodevelopmental delays [8].

We still do not fully understand the factors that determine the type, duration, and severity of symptoms caused by HCMV infections. The relationship between the infected host and the virus is likely to have significant implications. Current findings indicate the presence of a wide array of HCMV strains, highlighting the complexity of the virus’s behaviour [9]. Researchers have been trying to determine how these different strains might affect the course of infection and development of CMV disease. In this review, we explore the genetic variations present in HCMV strains, with a specific focus on their classification and the implications of these differences on the function of the virus and its associated diseases.

## 2. Landscape of Genetic Diversity

HCMV is a double-stranded DNA virus (dsDNA) and one of the longest human viruses (235–250 kb), with at least 170 canonical open reading frames [10,11]. It presents a standard herpesvirus class E genome architecture, with two unique regions (unique long UL and unique short US) that are flanked by a pair of inverted repeats (terminal/internal repeat long TRL/IRL and internal/terminal repeat short IRS/TRS), yielding the TRL-UL-IRL-IRS-US-TRS configuration [10,12,13].

### 2.1. HCMV Variability Is Mostly Found in “Islands” of Diversity

Genetic differences amongst HCMV strains were found over the entire genome, but some regions show a greater variability between strains compared to the rest of the genome, and they are known as “hypervariable”. These variable regions did not show nucleotide changes randomly distributed throughout the sequence, but they strongly clustered into well-defined “genotypes”, which are stable during the infection [9,14,15]. A genotype is defined as an individual’s DNA sequence patterns in a given region or gene, and an allele is one of the possible versions of a DNA sequence [16]. Hypervariable regions with multiple genotypes are also defined as “multiallelic”, as more than one allele is present. On the contrary, regions with only one allele are defined as “mono-allelic” rather than conserved, to consider the fact that variants can occur outside the hypervariable regions (Figure 1).

The genes used for genotyping are shown in Figure 2. Some of these genes encode glycoproteins essential for the viral life cycle, such as glycoproteins gB (*UL55*), gH (*UL75*), gN (*UL73*), and gO (*UL74*). Other highly variable genes used for genotyping are interesting because they encode human cellular homologues, such as *UL146*, which encodes a viral CXCL chemokine, and the *UL144* gene, which encodes a TNF-receptor homologue. Members of the *RL11* gene family have also been extensively studied because they exhibit considerable variability across different strains. Notably, one member of this family, *RL13*, is highly prone to rapid mutation during in vitro culture [17]. Most genotyping studies have focussed on a few highly variable genes identified through polymerase chain reaction (PCR)-based genotyping [18,19,20]. Although whole-genome studies are becoming more popular, initial attempts involved isolating the virus in cell culture, making it more susceptible to gene loss and mutations. In addition, researchers have relied on the direct sequencing of PCR amplicons from clinical samples, which could introduce artefacts, due to the high number of PCR cycles [21,22,23,24,25]. Recent studies have overcome these limitations by using target enrichment to facilitate the direct sequencing of strains found in clinical samples [22,26,27]. Another issue in genotyping studies is the identification of genotypes or alleles. Genotyping for candidate genes has often been carried out visually and/or with phylogenetic trees, using a limited number of sequences [18,28,29,30].

Suárez et al. recently introduced a new method that addresses some of the previous limitations [31]. They developed genotype-specific motifs from 163 HCMV sequences and confirmed their validity in 243 *UL73* and *UL74* genomes. This approach was later extended to include ten other hypervariable genes (*RL5A*, *RL6*, *RL12*, *RL13*, *UL1*, *UL9*, *UL11*, *UL120*, *UL146*, and *UL139*). Genotyping is accomplished by tallying reads that contain motifs specific to the genotypes of hypervariable genes. Although this method stands out for its use of a large number of genomes and a standardised approach to genotype generation, it has only focussed on a small number of candidate genes identified as hypervariable, and is further limited by its consideration of the whole gene boundaries, rather than the actual hypervariable region within the gene, as the units of variability and the definition of alleles by eye.

To better identify the boundaries of hypervariable regions and the number of alleles for each region, we utilised Hidden Markov Model (HMM) clustering within a comprehensive dataset of 253 HCMV genomes [32]. HMM can be used to determine the optimal number of sequence clusters or alleles that account for diversity across CMV genomes. Using an unbiased and probabilistic assignment model, our approach accurately identifies the nucleotide positions of multi-allelic and mono-allelic regions. Using this method, we described 74 multi-allelic regions with two to eight alleles each, comprising 14% of the genome, some of which were previously identified as hypervariable, but over 40 regions were novel [32]. These hypervariable regions were more evenly spread across the genome, providing more granular genotyping information.

### 2.2. CMV Hypervariable Genes

Previous analyses have mostly focussed on variable genes that encode four envelope glycoproteins essential for attachment, cell-to-cell spread, and interaction with the host’s immune system: glycoproteins B, H, N, and O [15,33,34]. Owing to their crucial role in initiating signalling transduction cascades in target cells and propagating HCMV infection, glycoproteins have been identified as key HCMV vaccine targets. Interestingly, these glycoproteins often form complexes, facilitating HCMV infection; for example, the glycoprotein M/glycoprotein N (gM/gN) dimer [35], the glycoprotein H/glycoprotein L/glycoprotein O (gH/gL/gO) trimer [36], and the gH/gL/gO/UL128-130 pentameric complex [37].

Three other variable genes were identified in the UL/b region. Two are human homologues; the *UL146* gene encodes a viral CXCL chemokine [38] and the *UL144* gene encodes a TNF-receptor homologue [39]. Less studied, but in the same genomic region, *UL139* has also been reported to be variable and is predicted to encode a membrane protein [15].

Another subset of variable genes includes the *RL11* gene family, whose members encode known or putative glycoproteins [10]. *RL13*, a member of this family of genes, is known to mutate rapidly in vitro [17]. The apparent ease with which *RL13* mutants are selected during cell culture raises the possibility that these mutations pre-existed in the clinical sample, potentially reflecting an expanded cell tropism in vivo [40].

#### 2.2.1. Glycoprotein B

gB is an HCMV envelope glycoprotein encoded by *UL55*, consisting of 907 amino acids (NCBI accession number: YP_081514). Five genotypes have been identified based on variations within the C-terminus, N-terminus, and gp55 cleavage site [28,41]. All five genotypes have been identified in different continents (Asia, Europe, and North America); however, their geographic distributions differ, with gB-1, gB-2, and gB-3 being the most prevalent genotypes in Europe [34,42,43]. In our study of HCMV diversity, we identified three multi-allelic regions (regions 22, 23, and 24, Table 1 and Table 2). The first two corresponded to regions identified at the protein level. We also identified a novel region (22) that is conserved at the protein level but shows several changes in the nucleotide alignment.

When combined, the multi-allelic regions form 12 haplotypes, some more frequently than others (Appendix A). Interestingly, alleles in region 24, which overlap with the antigenic AD2 region, show geographical segregation between European and African sequences [32].

Comparisons between previously identified genotypes and our haplotypes identified using HMM are shown in Table 3 and Appendix A. The phylogenetic tree (Appendix A), built with the protein alignment of representative sequences of different haplotypes and genotypes, showed the five clusters used to define the genotypes. However, it also revealed heterogeneity in some clusters, specifically in those identified as gB-2 and gB-4. Haplotype H2 clustered with gB-2, but it was also the closest cluster for H4, even though it had a different allele at region 24 (Appendix A). H1 and H7 clustered closely with gB-4 despite still showing some amino acid differences. The current genotyping system also does not capture H3 and H9, which cluster closer to gB-4 than other genotypes; however, they have a unique combination of alleles in regions 23 and 24. H5 (Merlin strain) and H6 (Towne strain) were classified as gB-1 and differed in region 22 of our analysis. Region 22 harboured many nucleotide changes, but was conserved in the protein alignments used to define the genotypes.

Many studies have investigated the possible link between gB genotypes and function. gB plays a critical role in CMV infection by mediating the final fusion event between viral and cellular membranes. gB fusogenic function is essential for viral entry and establishing infection [44,45]. Critically, gB is not only essential for viral entry but also for the cell-to-cell spread of the virus [44]. How the mechanisms of virus-to-cell entry and cell-to-cell fusion differ is unclear; however, several publications have reported gB-specific antibodies that are able to block one and not the other [46]. Samples collected from multiple body sites of the same individual showed different gB genotypes. One explanation is the presence of multiple HCMV strains in one host, due to a mixed infection. The partitioning of different gB genotypes into different body compartments may suggest tropism for different cell types [24,47].

Genetic variations in the gB sequences should be considered when evaluating vaccine efficacy against primary infection, reinfection, or reactivation. The gB-mF59 vaccine is based on the Towne strain, which has gB-1. There is some evidence that women immunised with gB-mF59 have better protection against primary infection with natural strains containing gB-1 compared to with viruses with other alleles [48]. This, along with the finding that the conserved region of Towne gB predominantly carries Africa-segregating SNPs, emphasises the need to test vaccines based on this strain for cross-protective immunity against European strains.

#### 2.2.2. Glycoprotein N

*UL73* is a polymorphic locus that encodes viral glycoprotein gN. gN is a type 1 transmembrane protein composed of 135 amino acids (NCBI accession number: YP_081521). Four genotypes (gN 1–4) were identified based on the differences in the N-terminal region (codons 1–87). Two subtypes of gN-3 (gN-3a and gN-3b) and three subtypes of gN-4 (gN-4a, gN-4b, and gN-4c) were identified [49]. These four genotypes are widespread and have similar distributions in Europe, Asia, and North America, except for gN-2, which is primarily found in North America [49]. We identified a multi-allelic region (region 28) that overlapped with *UL73*. Interestingly, this region (including 1990 nucleotides) overlaps with *UL73* and *UL74*, probably because of the high linkage disequilibrium between the two genes. For region 28, we identified seven alleles corresponding to the previously identified genotypes and subtypes (Table 4, Appendix A).

Studies investigating the role of different genotypes on gN function have focussed on humoral immunity to identify strain-specific neutralising activity. Indeed, anti-gM/gN dimer antibodies showed different activities against the AD169, Toledo, and TR strains [50]. This was confirmed in recombinant virus studies, in which four different gN genotypes were reconstructed in the AD169 virus backbone. Viruses with different genotypes were neutralised differently, suggesting that variability in gN could contribute to the evasion of an efficient neutralising antibody response [51,52].

#### 2.2.3. Glycoprotein O

Envelope glycoprotein O (gO) is a soluble protein encoded by *UL74* and is an essential component of the gH/gL/gO trimer. This complex is required for HCMV entry into host cells [53]. gO comprises 457–472 amino acids (NCBI accession number: YP_081522.1) depending on the number of strain-specific deletions. The main variable region is found at codons 1–98 (overlapping with the N-terminal region and where deletions might occur) [54], and minor variation is found at codons 270–313 [34]. This variation has led to the identification of five gO genotypes (gO-1 to gO-5) in HIV-positive and HIV-negative immunocompromised patients [54] and renal transplant recipients [55]. Further analysis identified subtypes of gO-1 (gO-1a, gO-1b, and gO-1c) and gO-2 (gO-2a and gO-2b) genotypes [20,56]. All five genotypes have been identified in Europe, North America, and East Asia. Recently, we identified seven alleles in European sequences, four of which were also found in African genomes, with a significant difference in distribution [32]. Our allele classification reflected the different genotypes and subtypes identified in previous studies (Table 5, Appendix A). The only difference was that the subtypes gO-1a and gO-1c were both classified as A1 using our method. This could be due to the selection of sequences used for our HMM methods, where all wild-type strains available in GenBank were included, but not all lab-passaged strains were included.

Our study observed that the gO and gN genes are located within the same multi-allelic region, designated as region 28. This co-location can be attributed to their close spatial proximity and the strong degree of linkage disequilibrium between the two genes [23,57].

Several studies have investigated the association between different gO genotypes and function. HCMV recombinant studies, in which different gO genotypes were reconstructed into the TB40E HCMV strain backbone, have shown that some genotypes elicit increased tropism for epithelial cells [58,59]. They also suggested that variability in gO can have a dramatic impact on cell-free and cell-to-cell spread, as well as antibody neutralisation [60].

#### 2.2.4. Glycoprotein H

Another key HCMV glycoprotein with substantial genetic diversity is glycoprotein H, which is encoded by *UL75*. gH is an essential component of the gH/gL/gO trimer and is part of the pentameric complex. In addition to gB, the pentameric complex is a vaccine target because of its role in entering epithelial, endothelial, and monocytic cells [34,61].

gH is 742 amino acids long (NCBI accession number: YP_081523.1) and is less variable than the other glycoproteins described. Two genotypes have been identified based on the N-terminal region (codon 1–37), both of which are found in Asia, Europe, and North America [34]. Although *UL75* is typically regarded as less variable than other glycoproteins, our research revealed the presence of three multi-allelic regions within the gene (regions 29, 30, and 31) (Table 1, Table 6 and Table 7).

Two alleles were identified in similar proportions in the European and African sequences of each region (Appendix A) [32]. The two genotypes, gH-1 and gH-2, partially overlapped with our two alleles in each region (Appendix A). However, after combining these three regions, we obtained six haplotypes (H1–H6) (Appendix A), probably because of recombination. Because two of these regions (regions 30 and 31) produce changes in the protein, this data analysis method allows us to assess the functionality of the interactions between different alleles.

gH is considered to be one of the main antigens for eliciting neutralising antibody responses [62]. This response appears to be strain-specific in fibroblasts and epithelial cells [63]. Cui et al. have shown that sequence SNPs within residues 27–48 (region 31) govern both the binding and neutralisation of virus entry into epithelial cells and fibroblasts. In addition, it has also been reported that the T-helper cell response to gH is strain-specific (aa 284–302, overlapping with our region 30) [64].

### 2.3. Viral Cytokine/Chemokine Proteins (Human Cellular Homologues)

#### 2.3.1. *UL144*

*UL144* is a tumour necrosis factor-α (TNF-α)-like receptor gene [10,65].

The sequence variability of *UL144* was initially reported in 45 low-passage HCMV clinical samples from congenitally infected infants, where three major genotypes were identified [65]. These three genotypes were confirmed by further analyses [14,66,67]. Our study identified a multi-allelic region overlapping *UL144* (4–107 codons) with three alleles corresponding to the genotypic groups identified previously (Table 8, Appendix A) [32]. The three alleles are present in Europe, America, and Africa, although they have slightly different prevalence rates (Table 8, Appendix A).

*UL144* likely plays multiple roles in regulating immunity to HCMV infection.

Molecular mimicry of cytokines and cytokine receptors is a strategy HCMV uses to modulate host immunity. The *UL144* gene, found in the UL/b’ region of the HCMV genome, has amino acid sequence similarity with members of the tumour necrosis factor receptor superfamily [68], and helps HCMV to evade the host immune system by inhibiting T cell activation, by binding to the B and T lymphocytes attenuator (BTLA) [69]. UL144 is a potent activator of NF-κB via a TRAF6-dependent mechanism. This activation enhances the expression of the chemokine CL22 through NFκB-responsive elements found in its promoter [70]. In addition, UL144 can also be anti-inflammatory by evading the CD160-mediated activation of NK cells [71].

Extensively passaged laboratory strains lack the UL/b’ region and do not encode UL144. The genes in this specific region (designated ULb’) are not deemed necessary for growth within fibroblast cell cultures [72].

#### 2.3.2. *UL146* and *UL147*

HCMV encodes two genes, *UL146* and *UL147*, whose protein products (vCXCL-1 and vCXCL-2) exhibit limited identity with CXC-chemokines [38]. Both genes show a consistently extreme variability of over 60% [29,66] at the amino acid level. Fourteen distinct *UL146* and *UL147* genotypes were identified (G1–G14) based on the variation throughout *UL146* and within a small region of *UL147* corresponding to a possible signal peptide [10,29]. Minor differences in genotypic frequencies have been identified among continents (Africa, Australia, Asia, Europe, and North America), but there has been no clear geographical separation [73]. Our study confirmed the high variability between strains, identifying eight alleles in a region overlapping *UL146* and *UL147*, with different frequencies between European and African sequences (Table 9, Appendix A). Our eight alleles corresponded to the eight most common genotypes (2, 5, 7, 8, 9, 11, 12, and 13) (Table 9, Appendix A). The absence of genotypes 1, 3, 4, 6, and 10 in our analysis is due to the unavailability of complete genome sequences for these genotypes in GenBank. Most studies describing these genotypes focus on sequencing specific genes of interest rather than the entire genomes. Since our method relies on whole-genome sequences, these genotypes could not be included.

Although both proteins are potential homologues for CXC chemokines, only the functional effects of pUL146 (vCXCL-1) have been reported in detail. Even though there is limited homology with the host, the UL146 acts as a functional CXC chemokine that binds to CXCR1 and CXCR2, and induces neutrophil chemotaxis and calcium mobilisation [38,74,75]. Disruption/deletion of the *UL146* gene from high-passage lab strains limited the ability of HCMV-infected fibroblast to promote neutrophil chemotaxis [74]. Early vaccine trials suggested the role of xCXCL-1 in infections in vivo, where the Toledo strain was found to be more virulent than the Towne strain [76].

### 2.4. Other Hypervariable Regions

Several other genes also have multiple alleles. Table 1 presents a complete list of all the multi-allelic regions and genes. *UL139*, for example, has been described in several studies and three to eight alleles have been identified [31,32,73,77]. The protein encoded by each HCMV *UL139* genotype contained a putative signal peptide sequence and a transmembrane region. A limited region of sequence identity (15 amino acids) has been identified between HCMV UL139 and human CD24, a highly glycosylated protein involved in B cell activation that is overexpressed in cancer [78,79]. While this raises the possibility that *UL139* may encode a CD24 homologue, the evidence remains preliminary, and further research is required to substantiate this claim.

Another subset of hypervariable genes is included in the *RL11* domain. The *RL11* gene family includes *RL11-RL13*, *UL1*, *UL4–11*, *RL6*, and *RL5A*, located near the genome’s N-terminus. Previous studies have identified two to five genotypes for some of these genes, with *UL1* exhibiting the highest degree of variation [32,80]. These genes are believed to encode putative transmembrane glycoproteins that are not essential for viral growth in cell culture. They are also absent in murine CMV [81] and *UL1* is absent in chimpanzee CMV [82]. A recent study looking at *RL11* evolutionary history confirmed that these genes are unique to Old World monkeys and Great Apes CMVs, and suggested that some human CMV-specific *RL11* genes emerged before the divergence of humans and chimpanzees, but were subsequently lost in the latter [83].

Some of the multi-allelic regions identified in our study were novel (n = 49) (Appendix A). Of these, 26 showed alleles with a different geographical prevalence, such as tegument proteins (*UL48* and *UL82*), capsid proteins (*UL86, UL48A*, and *UL80*), and envelope glycoproteins (*UL37* and *UL100*). The rest showed the same proportion of alleles in European and African sequences and included several membrane and envelope glycoproteins (*UL18*, *UL33*, *UL132*, *UL142*, and *US7*), tegument proteins (that is, *UL25*, *UL36*, *IRS1*, and *TRS1*), and membrane proteins (that is, *US14* and *US17*). Interestingly, five multi-allelic regions were identified in genes with no well-defined functions (*UL27*, *UL41/UL42*, *UL116*, *UL133*, and *UL148A/UL150A*), which contain potential transmembrane domains or signal peptides, and five non-coding regions overlapping with repeats (TATA box) and regulatory RNAs (Appendix A).

### 2.5. Outside the Hypervariable Genes

Although SNPs are typically located in multi-allelic regions, some can be found elsewhere in the genome. We identified several SNPs (n = 440) in the mono-allelic portion of HCMV, which showed geographical segregation between European and African sequences. Only 15% of these SNPs were non-synonymous. Of these, 16% overlapped with known B and T cell epitopes, providing little evidence that the geographic population structure in CMV, unlike EBV [84], is driven by a unique host immune pressure [32].

Other types of SNPs outside multi-allelic regions can appear during infection in a host. For example, administering antiviral drugs to treat HCMV viraemia disrupts the viral population, leading to the selection of drug-resistant variants [85]. Recent studies by our group and others [86,87] have shown that antiviral resistance variants can be found in transplant recipients with CMV viremia. These variants are especially prevalent in DNA polymerase *UL54* and protein kinase *UL97*, which are the primary targets for common antiviral drugs such as ganciclovir, cidofovir, and foscarnet [88,89]. Additional drug targets have been identified in *UL51*, *UL56*, and *UL89* for letermovir and *UL27* for maribavir [88,90,91] (our summary of genes with antiviral resistance variants can be found here http://cmv-resistance.ucl.ac.uk/herpesdrg/, accessed on 26 July 2024). All but one gene was found in the mono-allelic regions. *UL27*, an HCMV gene of unknown function that confers low-level resistance to maribavir [92], is the only gene with drug-resistance variants overlapping with a small multi-allelic region of 275 nt (nucleotides) (*UL27* is 1827 nt in Merlin strain NC_006273.2) (Table 1) [32].

### 2.6. Clinical Mutants with Nonfunctional Genes (Pseudogenes)

The genomes of some HCMV strains exhibit disruptions in their open reading frames (ORFs), which can lead to “pseudogenes”. This results from mutations that cause premature translational termination, such as SNPs introducing in-frame stop codons, splice sites, or structural variations (insertions and deletions), leading to frameshifting or a loss of protein-coding regions. In contrast to significant gene loss in highly passaged lab strains, more subtle mutations leading to pseudogenes have also been found in strains isolated from clinical samples. Sijmons et al. [25] showed that 75% of clinical strains are not genetically intact, but contain disruptive mutations in a diverse set of 26 genes. Only one out of four clinical isolates has the complete set of intact genes, with the other isolates having one (33%), two (27%), three (13%), or four (3%) mutated genes. None of these 26 genes are essential for the growth of fibroblast cells. Interestingly, most overlapped with multi-allelic regions (Table 1), except for *US9*, *UL111A*, *UL128*, *US13*, *UL136*, *UL30*, *UL145*, *US6*, and *US12*. However, all but one of the strains used in this study were passaged in cell culture. Therefore, some mutations might be the artefact of culture adaptation. More recently, Suárez et al. demonstrated that the distribution of pseudogenes in 91 strains sequenced directly from clinical materials was similar to the previous study. The most frequently mutated genes were *UL9*, *RL5A*, *UL1*, and *RL6* (members of the *RL11* family); *US7* and *US9* (*US6* gene family); and *UL111A* (encoding viral interleukin 10) [27]. It is not clear what the impact of pseudogenes is on the phenotype. Some pseudogenes originate from genes involved in immune modulation, such as *UL111A*, *UL40*, and *UL9*. *UL111A* encodes a viral interleukin 10 homologue, cmvIL-10. cmvIL-10 can bind to the human IL-10 receptor and compete with the human IL-10 for binding sites, despite the two proteins being only 27% identical [75,93,94]. The UL40 protein in human cytomegalovirus (HCMV) plays a crucial role in modulating the immune response, particularly in evading natural killer (NK) cell-mediated cytotoxicity. It achieves this by interacting with the HLA-E molecule, which is a ligand for NK cell receptors [95,96]. Interestingly, certain *UL40* variants are associated with higher levels of viremia and can affect the proliferation and activation of NK cells differently, impacting the clinical outcomes in transplant recipients [97,98,99]. *UL9* was predicted to be an immunoglobulin-binding domain [33]. However, the sample size was limited, and its involvement in pathogenesis is still speculative. Further studies are required to investigate the presence of pseudogenes in HCMV samples from different types of individuals (immunocompetent and immunocompromised) and tissues. The timing of deletions and their evolution over time also need to be investigated.

### 2.7. Repeats

Another source of sequence variation is the heterogeneity in the copy number of adjacently repeated elements or tandem repeats (TRs). Short tandem repeats, also known as “microsatellites”, are patterns of short motifs consisting of one–six bases.

Several studies have found that TR variations may affect the functionality and pathogenicity of viruses [100,101,102]. TRs in HCMV have been previously described, where insertion and deletion polymorphisms can differentiate between different viral strains and can be used as epidemiological markers [102,103]. Many of these microsatellites are found in non-coding regions. However, some are also found in coding areas, promoters, and other functional DNA, such as oriLyt (origin of DNA replication) [104].

Studying these microsatellites can provide valuable insights into the genetic variability, viral evolution, and gene regulation of viruses. Further research is needed, and long-read sequencing will help to reconstruct repeat regions [105].

## 3. Within Host Diversity

Multiple infections with diverse HCMV genotypes (also referred to as “mixed infections”) are common and have been documented in various patient cohorts, including those with intact immune systems [9].

In immunocompetent individuals, the presence of mixed infections with multiple HCMV strains was first identified in women attending a sexually transmitted disease clinic, as reported by Chandler et al. [106]. Subsequent studies have confirmed the existence of multiple gB and gN genotypes in samples from other immunocompetent populations, such as adults analysed post mortem [107], healthy children [108,109], and seropositive healthy women [110,111]. These studies suggest that reinfection with different HCMV genotypes is common throughout a person’s lifetime. This phenomenon has significant implications for pregnant women, the risk of congenital infections (see “Clinical Significance of Multi-Allelic Regions” section), and vaccine development, highlighting the importance of understanding the frequency and impact of reinfection [112].

Mixed infections are extremely common in transplant recipients [56,86,87,113,114,115,116,117]. Several studies have revealed a wide frequency of mixed infections in these patients, ranging from 15% to 90% [9]. However, this difference is likely due to variations in the patient populations and the range of methods used for genotype analysis. Mixed infections have been associated with poor outcomes [118,119]. However, more recent studies using whole-genome sequencing have found no association between multiple-strain infection and particular virological or clinical features, including mortality [87,115].

Multiple strains in clinical samples significantly overestimate the HCMV genome variability within an individual host [24,47]. The detected variants primarily represent genetic differences between strains, rather than the evolution of a single strain within the host [86]. When considering mixed infections, several studies [86,87,114,116] have shown that in infections with a single strain, the HCMV genome is highly stable in patients over time and during different reactivation episodes.

Infections involving multiple HCMV strains may disrupt genome stability by providing opportunities for homologous recombination [86,114,120], which plays a significant role in generating CMV diversity. Recombination has been demonstrated in a laboratory setting when two HCMV strains infect the same cell and interact during replication. This interaction generates progeny, whose genomes consist of new haplotypes formed by a mix of genotypes/alleles obtained from both parental strains. This process leads to the creation of a greater variety of haplotypes of different genotypes [121,122]. The occurrence of recombination in vivo is also supported by CMV sequencing in immunocompromised patients [114,123,124].

Many genotyping methods have been used to detect multiple strains in a sample. Most of these methods are PCR-based, and assume that sequence diversity occurs in a limited number of alleles/genotypes [9]. However, these studies only focussed on specific genes and did not allow for the detection of low-abundance alleles. As previously discussed, Suarez et al. [31] developed a method that uses whole-genome sequencing to detect genotype-specific motifs in 12 hypervariable genes. These motifs can be used to detect infections with multiple strains. However, this method uses only 12 genes and does not allow for the reconstruction of the entire genome. Reconstructing the whole genome is helpful, as it identifies which genotyped regions are part of the same genome, thus providing the potential to study epistatic interactions. Therefore, we developed a HaROLD in our laboratory. This programme uses a probabilistic framework to reconstruct the genomes of mixed infections and performs validation with high accuracy, using simulated and real mixtures of HCMV genomes [125]. Using HaROLD to reconstruct the whole genome allows for the accurate identification of the sequences of different CMV viruses present in one sample. This can then be used to explore viral dynamics over time, and identify within-host recombination [87,114].

## 4. Clinical Significance of Multi-Allelic Regions

Finding evidence supporting the connections between different HCMV alleles and pathology has been challenging. Multiple studies have investigated the link between HCMV genotypes of different genes, the presence or absence of disease, the severity of clinical symptoms, and transmissibility [9,14,15,66]. This type of study is useful because it could help identify prognostic factors for the likelihood and severity of disease in clinical settings.

HCMV mainly affects people with weakened immune systems and newborns, and most studies have focussed on these two groups of patients. In healthy individuals, primary infections typically do not exhibit symptoms, or only result in mild infections. Consequently, there has been limited focus on understanding the relationship between different HCMV genotypes and the likelihood or seriousness of primary infections in healthy individuals [9].

Infections during pregnancy are more concerning. HCMV is the most common infectious cause of congenital, acquired disability, ranging from sensorineural hearing loss to severe neurocognitive impairment [126]. Primary maternal infection during pregnancy confers a 30–40% risk of transmission to the foetus [126]. However, maternal immunity to HCMV before gestation does not prevent transmission to the foetus, and even women with long-standing immunity to HCMV can shed and transmit the virus [112,127]. In contrast to other congenital infections, such as rubella, parvovirus, and toxoplasmosis, the highest rate of congenital HCMV infection is found in populations in which women of childbearing age have the highest prevalence of serological immunity to HCMV, such as in Africa, Asia, and South America [112]. The reactivation of HCMV, but also reinfection with antigenically distinct strains, is also possible in immunocompetent women [128] and has also been observed in rhesus macaque models (RhCMV), where macaques with robust pre-existing adaptive immunity could be readily reinfected with another wild type or lab strain [129].

Researchers have studied how certain HCMV genotypes are transmitted in the womb, with a focus on gB and *UL144*. In short, viruses transmitted from mother to child share the same genetic sequences, and placental transmission appears to be independent of a specific viral strain [66]. Several studies have shown that all gB and *UL144* genotypes can be transmitted from mothers to foetuses and that the distribution of genotypes in infants infected before and after birth is similar [67,130,131,132,133]. Similar results were found for other genes investigated, such as gN and *UL149* [133,134].

Models with RhCMV looking at gB and gL have been used to investigate CMV viral populations in intrauterine transmission [135]. Rhesus macaques were inoculated with a mixture of the three strains. One of the three strains in the inoculum was dominant in all maternal and foetal CMV samples. However, the viral populations were still diverse, with minor haplotypes related to the dominant strain. These were consistently detected within the samples’ maternal tissues at multiple time points, indicating persistence over time and transmission between different maternal compartments. Some maternal haplotypes were also present in foetal and maternal–foetal interface tissues, supporting the idea of a mother-to-foetus transmission bottleneck [135]. Multiple closely related haplotypes were also identified in a small study from our group in five HIV-infected mothers, with compartmentalisation of CMV populations between the cervix and breast milk [136]. Babies were initially infected with one strain, but they commonly acquired a different strain from breast milk. In congenitally infected infants, the viruses that passed from mother to baby were similar to the dominant strain in the cervix, but had specific genetic differences compared to the strains in the breast milk and cervix of mothers whose babies were infected after birth. These genetic differences were found in 19 genes, notably in members of the *RL11* family, *UL40*, *UL74*, and *US27/US26* [136].

The second question is whether HCMV genotypes are associated with symptomatology at birth and neurological sequelae in congenital HCMV infections. Studies aiming to identify a viral marker of CMV disease outcomes have caused more debate than reached a consensus [66]. Most studies have focussed on gB and *UL144*; however, the results are inconsistent. Several studies have analysed the connection between gB and symptoms at birth and neurological consequences, such as hearing loss. However, none of the gB genotypes reliably predicted congenital CMV outcomes. The few studies that have found such a correlation are contradictory. For example, the gB-3 genotype was more prevalent in congenitally infected Japanese babies, especially those with SNHL [133]. Recently, Dong et al. [137] found that gB-3 was associated with a higher risk of skin petechiae in 42 cCMV-symptomatic infants than in 140 babies with postnatal infection, but they did not find a correlation with hearing loss. Furthermore, Bale et al. found that gB-3 was the most common genotype in babies from Iowa and common in asymptomatic infections, and they found no correlation with neurodevelopmental outcomes [67]. The results for *UL144* are also controversial. Some studies have found associations between types A and C of *UL144* and symptoms and unfavourable outcomes of CMV infection [52,138]. However, in cCMV infants with hepatic involvement, type B is associated with higher levels of hepatic enzymes [139]. In contrast, other studies did not find any link between *UL144* types and disease [67,131,132].

The results of investigations of other glycoproteins indicate no correlation between gN and gH genotypes and disease [52,66]. In contrast, Pignatelli et al. [134] monitored 74 congenitally infected newborns for symptoms of CMV disease at birth and during long-term follow-up, and revealed that newborns with symptoms at birth, abnormal imaging results, and sequelae were associated with the gN-4 genotype. Additionally, the genotype gN-4 has been linked to chorioretinitis in 42 cCMV Chinese babies [137]. The same study also found a relationship between gH-1 and hearing loss, although this finding lacks the support of other studies.

Some studies have shown a possible connection between the *UL146*/*UL147* types (G5 and G7) and symptomatic congenital CMV infection. G1, on the other hand, is more common in children with CNS damage and hepatomegaly [140]. G1 and G13 have also been linked to higher levels of IgG and IgM, and increased levels of hepatic enzymes in babies with cCMV and hepatic involvement [139]. However, other studies have shown no association between *UL146*/*UL147* genotypes and clinical manifestations of congenital infections [12,141].

Various factors have hindered the establishment of a clear relationship between the HCMV alleles and disease development. First, most studies were based on limited descriptions of alleles and genotypes. For example, most studies have focussed on only one gene at a time without considering that, due to the high degree of recombination, there are many possible allelic combinations (haplotypes) [23]. However, the link between haplotypes and clinical outcomes has not been adequately investigated. More comprehensive descriptions of multi-allelic regions [31,32] will likely help in designing better studies to investigate the link between alleles and HCMV clinical manifestations. Investigating the entire genome has proven to be challenging, owing to the limited sample sizes of these studies [66], and the problem of multiple testing and collective effort will be needed to overcome this. Another factor is the relationship between HCMV strains and the genetic background of the host. For instance, the host’s HLA status could impact the recognition of specific peptides for HCMV alleles, potentially influencing an individual’s immunological response [63,109].

Previous studies have explored the potential correlation between specific HCMV genotypes, primarily gB genotypes, and the clinical manifestation or severity of disease in immunosuppressed patients. However, only a limited number of these investigations have suggested a possible link between certain gB genotypes and disease severity in transplant recipients or individuals with AIDS [136,137,138]. A more recent study of 59 patients treated with allogeneic hematopoietic stem cell transplant has found that specific gB genotypes can have beneficial (i.e., earlier engraftment) and adverse effects (i.e., shorter overall survival) on the transplant host. Further studies are needed to validate these findings.

## 5. Evolution of Diversity

Despite the significant variability among HCMV strains, it is remarkable that a limited number of alleles can be defined for each multi-allelic region. Although the alleles are highly diverse, there is a much lower level of genetic variation within individual alleles. In addition, HCMV sequences are stable over short timescales in patients with single infections, indicating that alleles do not change over time, even in immunocompromised patients with chronic infections [55,86,87,130]. This is also true for cell culture [29]. Taken together, these findings suggest that alleles have a long history, perhaps emerging during the evolution of populations of early humans or their predecessors [10,73].

Geographical segregation has been described for other human herpesviruses, such as herpes simplex virus (HSV-1) [126], varicella-zoster virus (VZV) [103], and Epstein–Barr virus (EBV) [31]. However, most HCMV studies have indicated genetic similarity within viral sequences, regardless of geographic location [9,23,25,117]. In contrast, our recent findings suggest a distinct geographical separation of CMV genomes between Europe and Africa [32]. Our data indicated that most geographically informative SNPs were in mono-allelic genomic portions, which were under purifying selection [25]. Unlike EBV, in which host immune selection appears to drive local viral adaptation to different human host populations [84], our study showed no evidence that selection within known immunogenic regions of the HCMV genome is the dominant driver of the observed genetic variability. Instead, we postulate that bottleneck events, such as founder viruses, are plausible explanations. Interestingly, 32 of the identified multi-allelic regions followed the same pattern as the mono-allelic regions.

The majority of multi-allelic regions (n = 42) showed no geographical segregation, but maintained the full allele palette across Europe and Africa. According to our analyses, multi-allelic regions that were not geographically segregated were significantly enriched for genes encoding immunomodulatory functions. Many of these variable regions correspond to those previously identified to be in strong linkage disequilibrium (LD) by Lassalle et al. [23] (Table 1). For example, region 6 shows no geographical segregation and contains the nonrecombinant haplotype *RL11D* block, whose members have been proven or predicted to be virion membrane glycoproteins [129]. Variability in this region might be crucial for CMV adaptation to different primate species [23].

Our data strongly support the idea that the CMV genome is the result of two distinct evolutionary forces. The first type, supported by geographical segments of the genome, is a founder event. These events are characterised by genetic drift occurring in separate viral populations as a result of human movement and migration. Second, multi-allelic regions with similar allele frequencies worldwide reflect negative frequency-dependent balancing selection, a form of adaptation that maintains pre-existing diversity in the face of genetic drift [127]. This is similar to what happens with the major histocompatibility complex in humans, where the maintenance of multiple alleles at certain frequencies contributes to the ability of the immune system to respond to pathogens [22].

However, the study of the evolutionary history and phylogeographic origins of HCMV is complicated by the pervasive genome-wide recombination that occurs in HCMV. This leads to a large variety of possible haplotypes and hybrid strains [9]. An intriguing example of a hybrid strain is represented by the lab strain Towne, with 79% of SNPs in the mono-allelic portions segregating with African sequences and the rest segregating with European [32]. Apart from specific multi-allelic regions in high local linkage disequilibrium, the rest of the genome recombines freely [23]. Recombination is mostly observed in the mono-allelic part of the genome, where a high degree of conservation is likely an effect of strong purifying selection. Purifying selection is common in viral pathogens, indicating long-standing environmental constraints [137]. Such constraints in HCMV may be represented by a long co-evolutionary history with its host [77]. Indeed, recombination is known to increase the effect of negative selection by unlinking selected sites from the genomic background, providing a means to achieve the levels of purifying selection required to maintain HCMV genome functionality [23].

## 6. Conclusions

This review summarises the knowledge regarding genetic diversity in HCMV. Our recent study identified 74 regions, providing a more comprehensive understanding of HCMV genetic variability compared to previous research. Notably, our study provided more granularity for certain regions (i.e., *UL55*) and uncovered additional regions of interest. These findings provide valuable insights into the evolution and geographical population structure of HCMV. However, to gain a more complete picture, future studies should incorporate sequences from diverse populations worldwide. The investigation of viral markers associated with CMV-related diseases has generated more debate than consensus in the field. To address this, future research should focus on well-controlled populations and standardise genotype/allele classification. These approaches will help to clarify the relationship between viral genetic diversity and disease outcomes, ultimately advancing our understanding of HCMV pathogenesis and potentially informing more effective diagnostic and therapeutic strategies.

## Figures and Tables

**Figure 1 pathogens-14-00050-f001:**
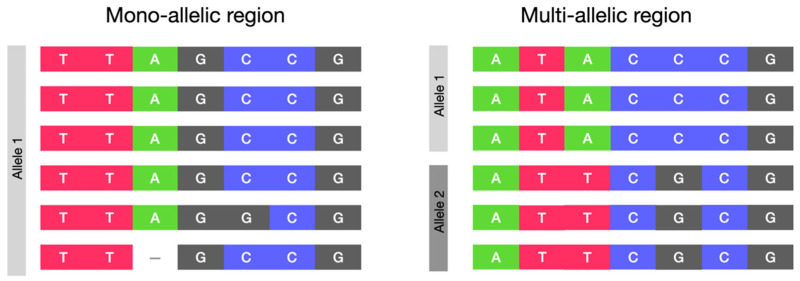
Schematic representation of mono-allelic and multi-allelic regions. Each row represents a different sequence.

**Figure 2 pathogens-14-00050-f002:**
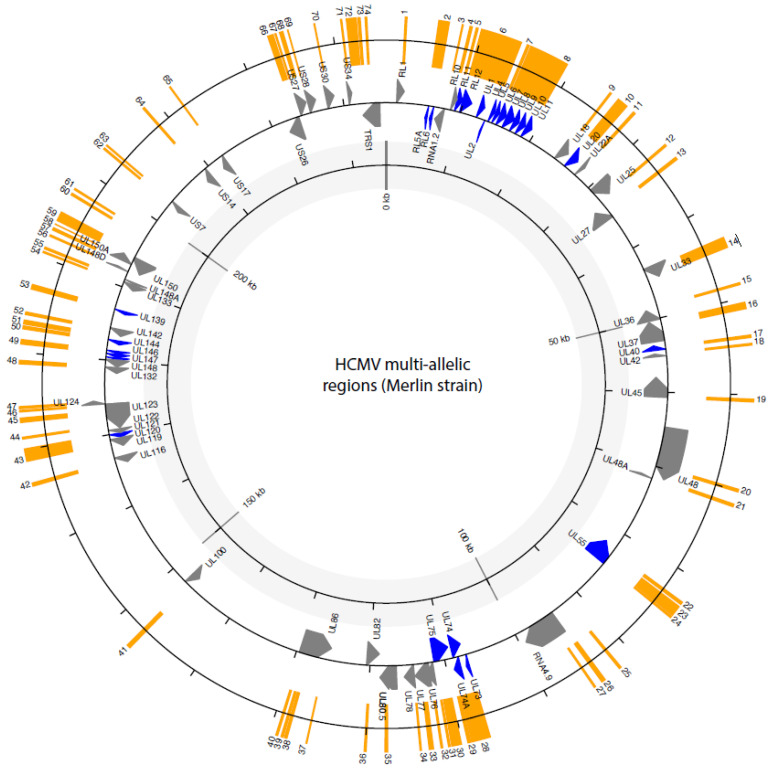
Summary of multi-allelic regions across the HCMV genome. The circle plot shows only the genes overlapping with the multi-allelic regions (in orange). In blue are genes previously used for genotyping.

**Table 1 pathogens-14-00050-t001:** Summary of multi-allelic regions identified in our study [32] previously used for genotyping. For each region, the table shows the number of alleles identified, open reading frames overlapping with the regions (genes), coordinates based on Merlin strain (NC_006273.2), geographic allele distribution for European and African sequences (this was calculated with Chi-square for independence and false discovery rate multiple testing correction, as described in [32]. Additional information has been added for linkage disequilibrium (LD), as found in [23]. A summary of previously identified variable regions overlapping with the multi-allelic region is provided: identified genotypes, most variable region of the genes identified, hypervariable regions overlapping with the multi-allelic region, as identified in [27] and geographic genotypes distribution if described). The deletion column is based on [25,27]. A summary of the possible link between HCMV alleles and functions is also provided.

Multi-Allelic Regions						LD	Genotypes Identified				Pseudogenes	Link Between HCMV Alleles and Function	
Region	Genes	Start	End	N of Alleles	Geographic Allele Distribution		Previous Identified Genotypes	Most Variable Region	Hypervariable Genotypes Regions Suarez et al., 2019	Geographic Genotypes Distribution		Phenotypes	Disease
1	*RL1*	1941	2121	2	No						Yes		
2	*RL5A RL6*	5387	6479	5	Yes	Yes			*RL5A*: 6; *RL6:* 7	Not described	Yes	Putative transmembrane glycoproteins; Not essential for viral growth in cell culture (Rawlinson, 1996); *RL13* gene mutates rapidly when HCMV wild-type strains are cultured in different cell culture systems (Stanton, 2010); Cell tropism (Stanton, 2010; Dolan, 2004).	
3	*RL9A*	7813	7914	2	No								
4	*RL10*	8620	8868	3	Yes								
5	*RL11*	9286	9479	2	No	Yes							
6	*RL11 RL12 RL13 UL1 UL2 UL4*	9840	14133	5	No	Yes	*UL1*: 3; *UL4*: 4 (Sekulin, 2007)		*RL12*: 10 (+subtypes); *RL13*: 10 (+subtypes); *UL1*: 10		*UL1, RL13, RL12*		
7	*UL5 UL6*	14765	14993	2	No	Yes	*UL6*: 4 (Sekulin, 2007)						
8	*UL10 UL11 UL6 UL7 UL8 UL9*	15163	19324	4	Yes	Yes	*UL7*: 3; *UL1*0: 3 (Sekulin, 2007)		*UL9*: 9; *UL11*: 7		*UL9, UL11*		
10	*UL20*	25622	26757	3	No	Yes			7				
17	*UL40 UL41A*	53875	54131	2	No			Region encoding the HLA-E-binding peptide (residues 15–23 in AD169) (Heatley, 2013)			Yes	Viral peptides derived from UL40 and presented on HLA-E are specifically recognised by the activating receptor NKG2C (Hamme, 2018); UL40 polymorphisms may aid evasion of NK cell immunosurveillance by modulating affinity of the interaction with CD94-NKG2 (Hartley 2013); NK cell response (Vietzen, 2021).	Not clear (Hartley 2013)
22	*UL55*	82720	83003	2	No	Yes	gB-5 genotypes (gB-1 to gB-5) (Wang, 2021)	Codons 26–70, gp55 cleavage site (codon 460)		All five genotypes have been detected in Asia, Europe, and North America; however, their distributions differ (Wang, 2022)		Essential role in the replication cycle of the virus. Required for virus entry and cell-to-cell spread of HCMV (Isaacson and Compton, 2009); Women immunised with gB-mF59 had better protection against primary infection with natural strains containing gB-1 compared to viruses with other alleles (Nelson, 2015, see also review Griffiths and Reeves, 2021).	Studies show inconsistent associations between gB genotypes and CMV disease severity or clinical manifestations (Pati, 2013; Yan, 2008; Tarrago, 2003). **cCMV Studies:** Research indicates no consistent link between gB genotypes and symptoms, sensorineural hearing loss, or neurodevelopmental outcomes in congenital CMV cases (Pati, 2013; Arav-Boger, 2002; Bale, 2000). Some studies suggest specific genotypes like gB-3 may be more prevalent, but findings vary (Yan et al., 2008; Dong et al., 2023). **Transmission and Clinical Outcomes:** Studies across HIV and transplant patients show gB genotypes do not correlate with clinical outcomes, though specific genotypes may be associated with complications in transplant recipients (Tarrago, 2003; Torok-Storb, 1997; Dieamant, 2013).
23	*UL55*	83278	84403	3	No								
24	*UL55*	84532	84716	3	Yes								
28	*UL73*	107059	109022	7	Yes		gN 4 genotypes gN-3 2 subtypes (Wang, 2021)	N terminal region	4 (+subtypes)	Not described		Humoral immunity (neutralising response) (Shimamura, 2006); Anti-gM/gN dimer antibodies possess differential neutralising activities against AD169, Toledo, and TR strains (Burkhardt, 2009; Pati, 2013).	Inconsistent findings (Arav-Boger, 2015) **cCMV Disease Studies**: No association found between gN genotypes and symptoms or sensorineural hearing loss (SNHL) in cCMV babies (Pati, 2013); Inconsistent results for gN-4 associated with chorioretinitis in symptomatic cCMV babies (Dong, 2023) and linked to symptoms at birth and sequelae (Pignatelli, 2010). **cCMV transmission:** All genotypes can be transmitted (Pignatelli, 2010). **Transplant Recipients:** No gN genotype was associated with a poorer outcome in solid organ transplant (SOT) recipients with CMV disease (Lisboa, 2012).
	*UL74*						5 (gO-1 to gO-5) + subtypes (Wang, 2021)	N-terminal region (codons 1–98), codons 270–313	5 (+ subtypes)	Differences in g) genotypes distribution in Japanese children vs. European samples (Wang, 2021; Yan, 2008)	Deletions at the N-terminus (in the first 90 aa) (Rasmussen, 2002)	gO-4 genotype showed an increasing tropism for epithelial cells vs. gO-1 genotype (Brait, 2020; Kalser, 2017) Different gO genotypes have an impact on neutralising antibody response to gH epitopes	
29	*UL75*	109129	109426	2	No	Yes	2	N-terminal region (codons 1–37)		Not described		gH is the main antigen eliciting a neutralising antibody response (Wang, 2021; Urban, 1996; Freed, 2013:) Cui, 2017: neutralising ability of certain gH-specific monoclonal antibodies was shown to be strain-specific in fibroblast and epithelial cells	**cCMV Disease Studies**: No association between gH genotypes and cCMV symptoms or SNHL in cCMV babies (Arav-Boger, 2015; Pati, 2013); gH-1 genotype was associated with hearing loss in symptomatic cCMV babies in Dong, 2023. **Transplant Recipients**: Renal transplant recipients with mismatched antibodies for gH had a higher incidence of acute transplant rejection and CMV disease (Ishibashi, 2007).
30	*UL75*	110100	111111	2	No								
31	*UL75*	111275	111445	2	No								
43	*UL119 UL120 UL121*	168817	170109	4	No	Yes			*UL120*: 4 (+subtypes)				
49	*UL146 UL147*	180852	181323	8	Yes		14 (G1–G14) Bradley, 2008; Dolan, 2004)		*UL146*: 14	Not described	Deleted in highly passaged lab strains (Cha, 1996)	Virulence (absent in attenuated strains; only in vivo), receptor binding affinity, signalling efficacy, chemotactic properties (Heo, 2008); HCMV UL146/UL47 are alpha-chemokine genes and share size and sequence similarity with human alpha-chemokines (Arav-Boger, 2005); vCXCL-1s differentially activate neutrophils and polymorphisms that affect the binding affinity, receptor usage, and differential peripheral blood neutrophil activation —> HCMV dissemination and pathogenesis (Ho, 2015).	**cCMV Disease Studies**: Inconclusive. No association between genotypes and disease (Arav-Boger, 2006; Berg, 2021); G1 more frequent in cCMV cases with CNS damage and hepatomegaly; G7 and G5 were predominant in postnatal CMV (pCMV) (Parawdoska, 2015); Linked UL146 genotypes G1 and G13 to higher levels of IgG and IgM antibodies, as well as elevated liver enzymes, in babies with cCMV and hepatic involvement (Guo, 2016).
50	*UL144*	182416	182725	3	No	Yes	3 (Arav-Boger, 2015)			No	Deleted in highly passaged lab strains (Cha, 1996)	Tumour necrosis factor alpha-like receptor; Role in vivo (Cha, 1996).	**cCMV Disease Studies:** Controversial Findings (Arav-Boger, 2015). Some associations with UL144 type C and symptoms (Pati, 2013) and type A and C with poor outcome in cCMV babies (Arav-Boger, 2002). Another study linked type B to higher enzyme levels in cCMV babies with liver involvement (Guo, 2016); No association between UL144 types and cCMV disease (Nijman, 2014; Bale, 2001; Picone, 2004). **cCMV Transmission Studies:** All genotypes can be transmitted (Yan, 2008; Bale, 2001; Revello, 2008, Picone, 2004).
53	*UL139*	186573	187057	4	No		3-8 (Qi, 2006; Bradley, 2008)	N-terminal portion (Bradley, 2008)	8 (+subtypes)	Not clear (Bradley 2008)	Deleted in highly passaged lab strains (Cha, 1996)	Shared sequence homology with human CD24 (signal transducer modulating B-cell activation responses). G1c contained a specific attachment site of prokaryotic membrane lipoprotein lipid (Qi, 2006).	
66	*US26 US27*	223336	223914	2	No	Yes	5				*US27*	Functional beta-chemokine receptor	No association with cCMV disease (Pati, 2013; Arav Boger, 2002)
67	*US27*	224108	224224	3	No								
68	*US27*	224607	224958	2	No								
69	*US27 US28*	225456	225513	4	No								

**Table 2 pathogens-14-00050-t002:** Multi-allelic regions in UL55.

Multi-Allelic Regions	Previously Identified Variable Regions in gB
*reg 24:* codons 24–85	codons 26–70
*reg 23:* codons 128–503	Codons 181–195; 311–317; gp55 cleavage site (codons 460).
*reg 22:* codons 595–689	Not identified.

**Table 3 pathogens-14-00050-t003:** Comparison between haplotypes identified in our study (H1–H10) and previously identified genotypes (gB1–gB5). H11 and H12 are not represented here because they were only identified in the reconstructed genomes from mixed infections, and protein sequences were unavailable. This table shows the GenBank accession identifiers for representative sequences for each haplotype H1–H10. Genotypes representative of protein sequences for gB1–gB5 were ACM48044.1, DAA00160.1, ADD39116.1, AAA45925, and AZB53144 [34].

Haplotype	Example Strain	Previously Identified Genotypes
H1	KY490079.1	gB-4
H2	FJ527563.1 AD169	gB-2
H3	KJ361956.1	gB-4
H4	KY490069.1	gB-2
H5	NC_006273.2 Merlin	gB-1
H6	FJ616285.1 Towne	gB-1
H7	KY490067.1	Separate cluster—closer to gB-4
H8	GU179289.1 VR1814	gB-3
H9	KY490088.1	Separate cluster—closer to gB-4
H10	KJ361971.1	gB-5

**Table 4 pathogens-14-00050-t004:** Comparison between the alleles identified in our study for region 28 (A1–A7) and previously identified genotypes for gN. The GenBank accession identifiers for representative sequences are shown in this table. Genotypes’ representative protein sequences were taken from this review [34]. The GenBank accession numbers of published reference genotype sequences are as follows: gN-1 (AD169 strain, P16795.1), gN-2 (Can 2 strain, AAL77763.1), gN-3a (PS strain, AAL77773.1), gN-3b (A8–27F strain, AAO24841.1), gN-4a (ZV strain, AAL77779.1), gN-4b (Towne strain, AGT36491.1), and gN-4c (Toledo strain, AAS48964.1).

Allele	Example Strain	Previously Identified Genotypes	Frequency in Europe	Frequency in America	Frequency in Africa
1	KY490061.1	gN-3a	20.5%	9.1%	23.3%
2	KY490065.1	gN-3b	8.4%	27.3%	3.3%
3	FJ616285.1 Towne	gN-4b	11.6%	18.2%	66.7%
4	NC_006273.2 Merlin	gN-4c	11.6%	0	0
5	KY490062.1	gN-4a	22.8%	0	0
6	FJ527563.1	gN-1	13.5%	9.1%	6.7%
7	KJ361956.1	gN-2	11.6%	36.4%	0

**Table 5 pathogens-14-00050-t005:** Comparison between the alleles identified in our study for region 28 and previously identified genotypes of gO. The GenBank accession identifiers for representative sequences are shown in this table. Genotypes representative of protein sequences were taken from this review [34]. GenBank accession numbers of published reference genotype sequences: gO-1a (AD169 strain, ACL51143.1), gO-1b (Cincy 2strain, ACS93309.1), gO-1c (Toledo strain, AAS48965.1), gO-2a (FUK19U strain, ABY48952.1), gO-2b (SW1102 strain, AAN40063.1), gO-3 (SW5 strain, AAN40074.1), gO-4 (Towne strain, AGT36493.1), and gO-5 (Merlin strain, YP_081522.1).

Allele	Example Strain	Previously Identified gO Genotypes	Frequency in Europe	Frequency in America	Frequency in Africa
1	KY490061.1	gO-1b	20.5%	9.1%	23.3%
2	KY490065.1	gO-2a	8.4%	27.3%	3.3%
3	FJ616285.1 Towne	gO-4	11.6%	18.2%	66.7%
4	NC_006273.2 Merlin	gO-5	11.6%	0	0
5	KY490062.1	gO-3	22.8%	0	0
6	FJ527563.1 AD169	gO-1a	13.5%	9.1%	6.7%
7	KJ361956.1	gO-2b	11.6%	36.4%	0

**Table 6 pathogens-14-00050-t006:** Multi-allelic regions in *UL75*.

Multi-Allelic Regions	Previously Identified Most Variable Regions in gH
*reg 31:* codons 3–60	codons 1–37
*reg 30:* codons 114–451	Not identified.
*reg 29:* codons 676–742	Not identified.

**Table 7 pathogens-14-00050-t007:** Comparison between haplotypes (H1–H6) identified in our study for regions 29, 30, and 31, and previously identified genotypes for gH (gH1 and gH2). The GenBank accession identifier for the representative sequences of each haplotype is shown in this table. Genotype representative protein sequences were taken from this review [34]. The GenBank accession numbers of published reference genotype sequences are gH-1 (AD169 strain, ACL51144.1) and gH-2 (Towne strain, AGT36494.1).

Allele	Strain	Genotype	Frequency in Europe	Frequency in America	Frequency in Africa
H1	KY490061.1	gH-1	26.05%	0	30%
H2	NC_006273.2 Merlin/FJ616285.1 Towne	gH-2	27.3%	40%	40%
H3	FJ527563.1 AD169	gH-1	20%	9.1%	13%
H4	JX512206.1	hybrid	1.86%	0	13%
H5	KJ361946.1	gH-2	10.70%	36.4%	0
H6	KP745640.1	hybrid	1.40%	27.3%	0

**Table 8 pathogens-14-00050-t008:** Comparison between alleles identified in our study for region 50 and previously identified genotypes for *UL144*. The GenBank accession identifiers for representative sequences are shown in this table. Genotypes’ representative protein sequences were taken from Lurain et al. [65]. The GenBank accession numbers of the published reference genotype sequences are as follows: group 1, AAF13363.1; group 2, AAF09111.1; and group 3, AAF09096.1.

Allele	Strain	Genotype	Frequency in Europe	Frequency in America	Frequency in Africa
A1	NC_006273.2 Merlin	Group 1	39.5%	9%	60%
A2	FJ616285.1 Towne/MF084224.1	Group 3	45.6%	55%	23.3%
A3	KY490064.1	Group 2	14.9%	36%	16.7%

**Table 9 pathogens-14-00050-t009:** Comparison between alleles identified in our study for region 49 and previously identified genotypes for *UL144*. The GenBank accession identifiers for representative sequences are shown in this table. Genotypes’ representative protein sequences were taken from Lurain et al. [29]. The GenBank accession numbers of published reference genotype sequences: 1 AAZ91734.1; 2 AAZ91735.1; 3 AAZ91718.1; 5 ABA02110.1; 7 AAZ91719.1; 8 AAZ91724.1; 9 AAZ91727.1; 10 ABA02161.1; 11 AAZ91729.1; 12 ABA02131.1; 13 ABA02122.1; and 14 ABA02092.1. Genotypes 4 and 6 have not been identified by Lurain et al. [29].

Allele	Strain	Genotype	Frequency in Europe	Frequency in America	Frequency in Africa
A1	KY490068.1	13	13.49%	27.3%	3.3%
A2	KY490084.1	7	2.33%	18.2%	3.3%
A3	MK290742.1	12	17.67%	9.1%	20%
A4	KY490088.1	9	12.09%	0	50%
A5	MK290743.1	5	13.95%	9.1%	13.3%
A6	MF084224.1	8	16.28%	9.1%	0
A7	NC_006273.2 Merlin	2	13.02%	18.2%	0
A8	KY490067.1	11	11.16%	9.1%	10%

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
