# Peer review of "Cytomegalovirus Genetic Diversity and Evolution: Insights into Genotypes and Their Role in Viral Pathogenesis"

_pathogens, 2025, doi:10.3390/pathogens14010050_

Round 1

Reviewer 1 Report

Comments and Suggestions for Authors

Venturini and Breuer have reviewed the literature around the impact of genetic diversity of human cytomegalovirus (HCMV) and how it impacts on viral pathogenesis. This is an important subject and  the review is timely. The review is generally well written with appropriate referencing of the literature, with a large supplemental file featuring frequencies of genotypes and phylogenetic trees. There are some weaknesses, such as making certain generalisations and a lack of detail/primary references in terms of the function of some of the HCMV genes discussed within the review.

Major comments:-

1. The authors state that the RL11 family are 'subjected to rapid mutations in cultured strains' (Page 5 line 129). To my knowledge and looking at Dargan et al., 2010, these mutations occur only in RL13 and are not reflective of the whole of the 14 genes of the RL11 family. This is also stated in Page 2, line 72 which refers to papers sequencing clinical strain (in which more RL11 family genes are mutated but in vivo), and Table 1. This generalisation should be corrected/clarified. The authors also state this rapid mutation of RL13 suggests its involvement in cellular tropism. This doesn't seem to be a strong argument and presumably there were other reasons for this statement in the original research.

2. Page 6, line 172 'Glycoprotein B is essential for HCMV's replication cycle of HCMV and is required for viral entry and cell-to-cell spread'. This sentence needs correcting. The exact function of gB in virus entry could be given.

3. UL144 is referred to as a 'truncated' TNFR receptor. It's not referred to as being 'truncated' in the original Benedict reference. It's closest homology is to HVEM and there are papers on the structure (Bitra et al J Biol Chem 2019) and function (Poole et al EMBOJ 2006 J Viral 2008, Cheung et al PNAS 2005 and Sedy et al, J Biol Chem 2017). The statement that is only found in clinical strains and low passage isolates also seems slightly misleading. Surely most HCMV strains are clinical/low passage and so most strains contain UL144, outnumbering the rare laboratory strains with deletions of/mutation in UL144.

4. UL146 and UL147 (Pg 10 line 314) do have homology with alpha-chemokine but this is fairly low. My understanding is that some chemokine activity has been shown for UL146 but not for UL147. It would be good to bring in functional studies to this section (e.g. Luttichau et al. J Biol Chem 2010, Berg et al, PLoS Pathogens 2022, Penfold et al PNAS 1999, Hey et al J. Immunology 2015 etc).

5. The authors state that 'The reason why the other five genotypes (1, 3, 4, 6, and 10) are missing from our analysis is that we could not find complete genome GenBank sequences that included these genoypes (Page 10 lines 325-326). This statement seems strange as presumable the sequence information was available for the definition of the genotypes and are other strains in these genotypes available. Can the authors clarify?

6. The authors state 'Interestingly, a region of sequence identity has been identified between HCMV UL139 and 348 human CD24, a highly glycosylated protein involved in B cell activation that is overexpressed in cancer (Altevogt et al., 2021; Fang et al., 2010)' (Page 11 line 347-349). The homology is very short approximately 15 amino acids. Many HCMV proteins will have this level of homology with a cellular protein, but it seems like a stretch to infer a functional relevance with this level of homology. 

7. This statement about the RL11 family 'They are also absent in murine CMV (Rawlinson et al., 1996) and UL1 is absent in chimpanzee CMV (Davison et al.,2003). These genes also mutate rapidly when wild-type HCMV strains are cultured in disinct cell culture systems (R. J. Stanton et al., 2010).  (Page 11 line 358-360)should be expanded to include Old World monkeys which also have the RL11 family and New World monkeys which do not. There is a recent bioinformatic analysis that could be included (Evolution of the Cytomegalovirus RL11 Gene Family in Old World monkeys and Great Apes by U Litvin et al. Virus Evolution 2024).

8. The use of term 'pseudogenes' to describle genes that are mutated either in culture or in clinical strains is possibly misleading. Most strains in vivo will carry a functional gene, the mutation in vitro is an adaption to cell culture in genes that are non-essential for virus growth in vitro but impair replication. The gene is functional in the original strain but becomes a pseudogene. Mutated genes in the clinical strains are presumably because of selection in the host in vivo potentially due to selection by the immune system. The majority of strains (70% plus) carry a functional gene so only a sub-populations are carrying what could be termed a 'pseudogene'. The lack of functional characterisation of these mutated genes also means whether they are non-functional or possibly carry an altered function isn't currently known. The authors quote a figure that 75% of clinical strains carry a pseudogene but presumably these are different genes in each strain so again the bulk of the viral 'population' carry a 'functional' intact copy.

9. References for UL111A and UL40 (Page 10 lines 420 and 425) should be added to. Suarez is a sequencing paper, there should be functional papers characterising UL111A. The signal peptide of UL40 has the function ascribed to it (up regulation of HLA-E which binds inhibitory NKG2A NK receptor but also activating NKG2C receptor). The original reference is Tomasec et al Science 2000 (as well as Ulbrecht et al  J immunology 2000 Carbon EJI 2001 Wang et al PNAS 2002) as well as the Prodhomme reference. It might be useful here to also discuss the expansions of NKG2C NK cells in HCMV seropositive individuals here.

10. The authors talk about a geographic effect on HCMV variation (Page 17 line 651-652). Can the authors clarify this as the general understanding to my knowledge is that HCMV does not vary geographically like for example EBV and that genotypic variation is very ancient?

11. Can the table and figure legends be expanded to help the reader for main manuscript and supplemental information? For Table 1, can functional detail be added in a gene specific manner?Table 1, UL40 is not a HLA-E homologue as stated but rather contains a signal peptide that is 'loaded' into HLA-E and leads to its up regulation on the cell surface of the infected cell. There it can bind to NKG2A or NKG2C with varying affinities (Hammer et al Nature Immunolgy 2018). Table 1, Can 'Pseudogene' be changed to 'Mutated genes'? Supplemental table 1, can an explanation of what geography means be added and an additional column with any published function of the gene product?

Minor points:-

1. Figure 2 has low resolution which should be improved for publication.

2. AD169 and Ad169 are used. Can AD169 be used throughout?

3. Use 'genotypes' instead of 'alleles' predominantly

4. Refer to gB throughout (rather than GB)

Reviewer 2 Report

Comments and Suggestions for Authors

This review manuscript titled “Cytomegalovirus Genetic Diversity and Evolution: Insights into Genotypes and Their Role in Viral Pathogenesis” by Venturini C and Breuer J  evaluates the genetic variations present in HCMV strains with a specific focus on their classification and the implications of these differences on the function of the virus and its associated diseases. The authors have considerable expertise in the studies of genomic and geographical structure of HCMV and recently published a research paper in PNSA about this subject. The manuscript is well written and well organized. I only have following minor concerns:

The authors should pay more attention to most recent studies/publications. Although some of recent studies are cited, following recent studies particularly about CMV drug resistance gene mutation are not described/cited in this review manuscript:

1. J Infect Dis 2024 Jun 10:jiae287. doi: 10.1093/infdis/jiae287. Cytomegalovirus antiviral resistance among participants in the phase 3 trial of letermovir vs valganciclovir prophylaxis in kidney transplant recipients

2. Pharmaceuticals 202417(4), 428; https://doi.org/10.3390/ph17040428

Variations in the viral UL55 locus imparting both beneficial (earlier platelet engraftment, less frequent MRD post HSCT) and adverse effects (shorter overall survival, more frequent acute GvHD, less frequent 100% chimerism at day 90) to the transplanted host.

3. Antiviral Res. 2022 Nov;207:105422. doi: 10.1016/j.antiviral.2022.105422.

Relative frequency of cytomegalovirus UL56 gene mutations detected in genotypic letermovir resistance testing.

4. Antiviral Res. 2024 Aug;228:105935. doi: 10.1016/j.antiviral.2024.105935. Epub 2024 Jun 14. PMID: 38880196

UL56 and UL89

5. Antiviral Res. 2024 Feb:222:105792. doi: 10.1016/j.antiviral.2023.105792. Epub 2023 Dec 30.

UL97

Comments on the Quality of English Language

The manuscript is well written.

Round 2

Reviewer 1 Report

Comments and Suggestions for Authors

The authors have dealt with the majority of my concerns. There are a few minor corrections needed in my opinion before publication.

1. Page 2, lines 72-73. Are the Puchhammer-Stöckl & 73 Görzer, 2011; Sijmons et al., 2014 the best primary references showing that RL13 mutates using in vitro culture systems? Dargan et al was used later on and would seem to be a more appropriate reference.

2. Page 10, lines 322-333. Can the authors add in the Cheung 2005 reference (PNAS doi: 10.1073/pnas.0506172102) in addition to the existing references as the earliest functional description as an immune evasion function targeted T-cell activation by targeting BTLA?

3. Page 11, lines 367-371. The homology between UL139 and CD24 is limited. Can the authors state that the homology is only in a 15 amino acid region? Overemphasising this could lead to readers being misdirected and the importance taken as fact through repetition.

4. Page 13, lines 444-446.  This section is slightly confusing. UL40 signal peptide has homology with the leader sequence/signal peptide of HLA-I. So this peptide is present in all cell expressing HLA-I (all nucleated cells). This is most relevant to the HCMV-infected target cells. The leader sequence of HLA-I is the natural peptide ligand for HLA-E and stabilises HLA-E allowing its transport to the cell surface. The signal peptide of UL40 substitutes for the HLA-I peptide in HCMV-infected cells . HLA-E binds to the paired receptors NKG2A (inhibitory) and NKG2C (activating) which are present on NK cells.

5. Table 1. One of the columns has been changed to 'deletions'. Are these actually deletions or disruption of the ORF by frame shift/nonsense mutations? UL40 it's a single peptide in the signal peptide and binds NKG2A and NKG2C was mentioned previously.
